# Multi-Node Small Radar Network Deployment Optimization in 3D Terrain

**DOI:** 10.3390/s25071964

**Published:** 2025-03-21

**Authors:** Zhiyi Wang, Min Wang, Xinghui Wu, Shuyuan Yang

**Affiliations:** 1National Laboratory of Radar Signal Processing, Xidian University, Xi’an 710071, China; w5z2y0@163.com (Z.W.); wuxingxinghui@stu.xidian.edu.cn (X.W.); 2Information Center, Xi’an University of Finance and Economics, Xi’an 710100, China; 3School of Artificial Intelligence, Xidian University, Xi’an 710126, China; syyang@xidian.edu.cn

**Keywords:** radar network, deployment optimization, radio wave propagation, layered effective coverage rate

## Abstract

When deploying multi-node small radar networks in cities or mountains, it is crucial to consider the influence of terrain. The propagation of radio waves in areas with known three-dimensional (3D) terrain differs significantly from that in free space. However, existing radar deployment optimization methods often rely on simplistic propagation models that do not accurately capture the variations in coverage at different heights. Therefore, the parabolic equation model (PEM) is first introduced to radar network deployment considering terrain constraints. After obtaining coverage results for different altitude layers, the Layered Effective Coverage Rate (LECR) is proposed as our optimization objective. Then, the nondominated sorting genetic algorithm III (NSGA-III) is employed to address this multi-objective optimization problem. Finally, the experimental results demonstrate the superiority of introducing PEM and the effectiveness of NSGA-III.

## 1. Introduction

Traditional radars are not effective in detecting low-altitude slow-moving small (LSS) targets and are not suitable for use in complex environments [1,2,3,4,5,6] such as cities and mountains. LSS targets, with unmanned aerial vehicles as typical representatives, have low flight altitudes, small RCS (radar cross-section), and slow speeds. In critical air defense scenarios, they can carry out low-altitude penetration by exploiting terrain shielding. Techniques in [3,4,5,6,7,8,9] such as multi-static radar system, MIMO radar system, and multifunctional radar networks can enhance detection performance. Another effective detection approach is to deploy a multi-node small radar network. When confronting LSS targets, small radars offer the advantages of low hardware costs, convenient setup, and swift maintenance. The deployment of a multi-node small radar network constitutes a crucial aspect of filling detection blind spots at low altitudes.

This paper focus on the optimization of multi-node small radar network deployment under terrain constraints (Figure 1). On the one hand, during the deployment of such a network, the radars are of the same type, which means that installation constraints and maintenance requirements can be negligible, and the erection costs remain relatively fixed. On the other hand, it is crucial to consider the influence of terrain constraints. Terrain data provide valuable information, including headspace blind areas (HBAs), beam-blocking effects, and multipath effects. However, there is currently almost no research on this issue.

The radar network deployment (RND) problem is a challenging problem and is recognized as an NP-hard problem. Related research can be categorized into three aspects: optimization objectives [1,2,4,5,7], evolutionary algorithms [6,9], and propagation models [4,10,11,12]. There are various optimization objectives, such as effective coverage rate (ECR) in [9], angle of arrival accuracy and time difference of arrival in [2], non-connected area constraints in [5], and interference constraints in [4]. To tackle this issue, several evolutionary algorithms (EAs) have been introduced, including the genetic algorithm (GA) presented in [13], particle swarm optimization (PSO) detailed in [4,10,11,14], the multi-objective evolutionary algorithm based on decomposition (MOEA/D) discussed in [15], and the nondominated sorting genetic algorithm II (NSGA-II) referenced in [13,16,17]. Some research [11,13,14,15,16,18] has gradually shifted its focus to 3D terrain, as reported in [19,20,21]. It is crucial to incorporate a propagation model into RND when terrain data are available. However, radio wave propagation in 3D terrain environments is complex and cannot be accurately modeled solely by free space (FS) in [4,10], line of sight (LoS) in [11,14,15,16], or path loss (PL) in [12].

The parabolic equation model (PEM) under terrain constraints has been widely investigated in [19,20,22]. Wang et al. studied the influence of terrain on single-radar deployment and emphasized that ground clutter determines the detection of LSS targets in urban environments. Although the computational complexity of PEM is higher than that of FS, LoS, and PL, PEM provides more guidance for practical radar network deployment. Therefore, it is important to use a more objective approach, as stated in [23], rather than relying solely on outdoor tests during deployment.

PEM provides a more objective approach to propagation modeling under terrain constraints, which offers the following benefits: (1) High-precision ground clutter prediction quantifies the differences in propagation losses among different deployment scenarios, avoiding unrealistic situations such as radar candidate sites located in valleys or on water surfaces. (2) By introducing the antenna pattern, a more precise range of HBA can be obtained, and the impact of the atmosphere and terrain on propagation can be described. These two factors help to more accurately predict the coverage of radar networks at different altitudes, preventing the maneuvering breakthrough of LLS targets.

The main contributions of this article are as follows:In this paper, PEM is innovatively used to optimize the RND. By utilizing known terrain data, we can obtain propagation losses of the 3D region of interest (ROI) and, further, joint detection probabilities of the 3D ROI.To consider the effective coverage of different altitude layers, we introduce a new Layered Effective Coverage Rate (LECR) as a part of the optimization objective. As this is a multi-objective optimization problem (MOP), we propose NSGA-III to maximize the coverage of each altitude layer.Additionally, our experimental results, based on the high-resolution DEM data of a Chinese city and open GIS data from http://www.webgis.com/terr_us75m.html (accessed on 29 May 2002), demonstrate the necessity of incorporating PEM and the generalization ability of the proposed method for terrain data.

The proposed method has two major advantages: (1) Compared to FS, Los, and PL, PEM provides more details in predicting propagation. (2) In a 3D environment, LECR can reflect coverage at different altitude levels, helping us analyze coverage blind spots. Additionally, simulation experimental results demonstrate the necessity of incorporating PEM in comparison to other propagation models and show that the proposed method is an effective and efficient method for solving practical LSS early warning radar network deployment problems.

This article is organized as follows. Section 2 presents the problem formulation and introduces the joint detection probability of a radar network. In Section 3, 3D PEM and a Split-step Fourier transform (SSFT) technique are introduced. The proposed method is described in Section 4, with effective coverage across different altitudes being the core of the proposed method. Section 5 describes the experimental validations and comparisons using various terrain data and experiments. Finally, Section 6 concludes this article.

## 2. Problem Formulation

A multi-node small radar network is typically composed of several small radars, with each radar serving as a node in the network and individually responsible for aerial surveillance tasks. When nodes communicate with the host computer, they do not transmit radar data but only transmit the detection results of individual radars. Small radars are usually characterized by their simple structure, low cost, and quick setup, making them highly suitable for low-altitude blind spot compensation tasks in strategic locations. The small radars within a network are usually of the same model and have similar detection performance. Therefore, although the deployment of small radars is less affected by terrain, the overall detection performance of a multi-node small radar network is mainly limited by the deployment scheme once the number of nodes is determined.

Assuming there are N radars in the radar network N, the joint detection probability is(1)$$P^{net}=1-\prod_{i=1}^{N}\left(1-P^{i}\right)$$
where $P^{i}$ is the detection probability of i-th radar $R_{i}$. Using a single pulse square detector, $P^{i}$ can be described by Marcum *Q*-function [24] as follows:(2)$$P^{i}=Q(S_{nr},T_{h})$$
where $S_{nr}$ is the signal-to-noise ratio (SNR) of the Cell Under Test (CUT), and $T_{h}$ is the threshold of the Constant False-Alarm Rate (CFAR) detector. $S_{nr}$ is the ratio of the received power $P_{r}$ to the system noise power, and $P_{r}$ can be calculated by the radar equation [25]:(3)$$P_{r}=\frac{P_{t}G^{2}\lambda^{2}}{{\left(4\pi \right)}^{3}R^{4}L_{s}}\cdot \sigma \cdot F^{4}$$
where $P_{t}$, G, λ, and $L_{s}$ are the transmission power, transmission and reception antenna gain, wavelength, and system loss of $R_{i}$, respectively, σ is the radar cross-section of the target, R is the distance between radar and target, and F is the propagation factor. Obviously, $P_{t}G^{2}\lambda^{2}/L_{s}$ is a constant for $R_{i}$, and $P_{r}\propto {\left(F/R\right)}^{4}$.

In a 3D surveillance region M={M,M⊂A}, the propagation factor $F^{M}$ at M can be calculated as follows:(4)$$F^{M}=L_{\mathrm{fs}}^{M}-L_{\mathrm{dB}}^{M}$$
where $L_{\mathrm{fs}}=20\log(4\pi R/\lambda )$ represents the free space propagation loss, and PEM can provide a more precise estimate of the propagation loss $L_{\mathrm{dB}}^{M}$ at M compared to FS, Los, and PL.

## 3. Propagation Loss

In [4,10], since terrain is not taken into consideration, $L_{\mathrm{dB}}=0$, and thus $F^{M}$ is determined by $L_{\mathrm{fs}}$. For the more complicated 3D terrain in [11,14,15,16], the occlusion of 3D terrain is modeled using LoS, which is a simplified model that describes terrain occlusion with binary 0/1. However, this method tends to have significant errors when used for radar deployment in complex terrain conditions. The coverage strength model in [26] considers criteria such as distance, field of view, occlusion, etc., and also employs a binary system. Propagation losses under 2D terrain constraints can be estimated well using PEM, as described in [20,22,27]. In [19], PEM was extended to 3D scenarios and achieved good experimental results. Therefore, in this paper, the wave propagation in M can be approximated using PEM and then $L_{\mathrm{dB}}$ solved by SSFT in 2D space, as demonstrated in [19,28]. Due to the high computational complexity of this solution, this step is performed offline and in parallel for each candidate site.

### 3.1. Split-Step Fourier Transform

We assume that the direction of wave propagation is along the positive r-axis, the x-axis indicates the height, and the field component $\psi (x,r)=e^{ikr}u(x,r)$. The standard parabolic wave equation is described as [22](5)$$\frac{\partial^{2}u}{\partial x^{2}}+2ik\frac{\partial u}{\partial r}+k^{2}\left(n^{2}-1\right)u=0$$
where k=2π/λ is the free-space wavenumber, n is the refractive index of the propagating medium, and u(x,r) is the reduced function.

The numerical solution of Equation (5) is given in [19,22] using the wide-angle Fourier Split-step approach:(6)$$u(r+\Delta r,x)=e^{\left[ik(n-1)\Delta r\right]}F^{-1}\left\{e^{-ik_{x}^{2}\frac{\Delta r}{k+\sqrt{k^{2}-k_{x}^{2}}}}F\{u(r,x)\}\right\}$$
where F and $F^{-1}$ represent the Fourier transform and inverse transform, $k_{x}$ is the component of k in the x direction, and Δr is the step length, respectively.

### 3.2. Three-Dimensional Propagation Loss

The wave propagation in 3D space M can be modeled using PEM and then $L_{\mathrm{dB}}$ solved by SSFT in 2D space, as in [19,22].

The azimuth angle is measured clockwise from the positive *y*-axis. The *z*-axis corresponds to the height above ground level. Similar to radar beam azimuth scanning, u(r,x,ϕ) can be calculated in A, where ϕ∈[0,360) represents the azimuth angle and $A=\left\{\left(x,y,z\right)\right\}$ is a Cartesian coordinate system space. Through coordinate transformation and linear transformation, as described in [19], the propagation loss $L_{\mathrm{dB}}^{M}$ at M can be calculated from u(r,x,ϕ). Here, the calculation process of the detection probabilities for all candidate sites is summarized in Algorithm 1.
**Algorithm 1** Summary of solving detection probabilities**Input:** A DEM/GIS data and radar parameters, such as wavelength, pattern, and beamwidth.
**Output:** The set
$\left\{P^{s},s=1,...,S\right\}$ of detection probabilities to each radar.  **for** 
s=1,...,S
**do**    **for** 
ϕ=0,...,360
**do**     **while** 
$z\leq R_{\max}$
**do**      Calculate the 2D propagation loss
$L_{\mathrm{dB}}^{\phi }$ using Equation (6)      
z←z+Δz
     **end while**
    **end for**    Converting
$L_{\mathrm{dB}}^{\phi }$ to
$L_{\mathrm{dB}}^{M}$ through coordinate transformation    Calculate
$P^{s}$ using Equations (2)–(4), and save it as data file.   **end for**


An example of the propagation loss $L_{\mathrm{dB}}^{M}$ in a known DEM is shown in Figure 2. The DEM data used are the same as in [19] and are used as input for Algorithm 1. The radar is positioned at the center of the map. It can be found that there is a blind area at the radar top that gradually expands with increasing altitude; the energy emitted by the radar spreads along the *x*-axis and forms a certain beam width; and the impact of ground objects such as rivers, mountains, tall buildings, etc., on the propagation can be observed in the ground area. Compared to FS, Los, and PL, PEM provides more details in predicting propagation.

### 3.3. Complexity Analysis

The computational complexity of both F and $F^{-1}$ in (4) is $o\left(L\log L\right)$. Therefore, a large complexity $o\left(SN_{a}N_{r}L\log L\right)$ is generated in the second step, where *S* is the number of candidate sites, $N_{a}$ is the number of azimuth beams, and $N_{r}$ is the number of steps corresponding to the maximum detection distance $R_{\max}$ of the radar. For simplicity, we assume that all radars are the same in our work. Due to the high computational complexity, this step is computed offline in parallel. In addition, the size of matrix $P^{s}$ is significantly larger than the size of the corresponding terrain data, both of which are 3D matrices, but the former contains information from more altitude levels. To reduce computation, the matrix $P^{s}$ corresponding to high-resolution terrain data can be down-sampled, even though this approach neglects the shadowing effects of some small geographical features.

In general, the computational complexity of PEM is higher than that of FS, LoS, and PL because $\log L>M_{d}>N_{p}$, as shown in Table 1, where $M_{d}$ is the average step size of the distance between the radar and the surveillance region, and $N_{p}$ is the number of path losses. However, introducing 3DPEM brings several benefits. First, it allows for more accurate modeling of radio wave propagation, and diffusion modeling is more accurate, taking into account radar parameters such as wavelength, distance, beam width, and antenna pattern. Second, the use of refined models for terrain and atmosphere results in more realistic predictions of propagation. Finally, after moving away from the binary 0/1 approach, errors in coverage, propagation factors, and propagation losses are reduced.

It is important to note that this approach also highlights significant differences in propagation at different altitudes. As the altitude decreases, the impact of ground clutter becomes greater, and radar networks must take measures to prevent LLS targets from exploiting strong ground clutter for penetration. On the other hand, as the altitude increases, the complementarity between blind zones at the top of radars should not be overlooked. These considerations are crucial for the deployment of radar networks in complex terrain.

## 4. The Proposed Method

### 4.1. Layered Effective Coverage Rate

After obtaining the numerical solution of PEM, we obtain $P^{i}$ by Algorithm 1 and $P^{net}$ by Equation (1). In [10], the effective coverage region A(Θ) is defined as the region in which $P^{net}$ of each resolution cell is higher than a required threshold. The same operation is also adopted in this paper.

ECR [29] for the surveillance region M can be described as(7)$$C(\Theta )=\frac{A(\Theta )\cap M}{M}\times 100\%$$
where $\Theta =\{\theta_{i}={\left(x_{i},y_{i},z_{i}\right)}^{T},i=1,...,N\}$ is the position set of radars, $A(\theta_{i})\subset A$ represents the corresponding effective coverage region of $R_{i}$, and T represents the transpose operation. Since the effective coverage area A(Θ)∩M is a set in 3D space, there are different blind areas and terrain blockages at different altitude levels, as shown in Figure 3. Therefore, we introduce a new Layered Effective Coverage Rate (LECR) $C_{L}=\left\{c_{l},l=1,...,L\right\}$, which is a set of ECRs for all horizontal planes $\left\{(x_{l},y,z)\right\}$ in A with an altitude of $x_{l}$, l=1,...,L. Then, the initial deployment optimization problem aims to maximize the effective coverage rate of the entire surveillance region M, which can be formulated as(8)$${\hat{\Theta }}_{1}=\arg\max\sum_{M}c(\Theta )=\arg\max\sum_{l=1}^{L}c_{l}(\Theta )$$
where $c_{l}$ is the effective coverage rate of the radar network N in the l-th elevation plane.

The effective coverage area of the radar in Figure 2 can be determined from the propagation loss by Equations (2)–(4). Without considering the terrain, in other words, if the radar is positioned on a 2D plane, its effective coverage rate in that plane is determined by its maximum detection radius. In a 3001 × 3001 m^2^ surveillance area, when the radar’s maximum detection distance is 1500 m, the effective coverage rate is approximately 78.5%, as shown in Figure 3. However, when considering the impact of terrain, we find that the coverage of different altitude layers is significantly different. Therefore, it is necessary to represent the effective coverage of ROI by layered effective coverage.

It is expected that all $c_{l}$ values are at their maximum at each l and are balanced. Since maximizing layered effective coverage (LEC) involves maximizing the ECRs across L altitude layers, maximizing LEC is an MOP, expressed as(9)$${\hat{\Theta }}_{2}=\arg\max C(\Theta )=\left(c_{1}(\Theta ),c_{2}(\Theta ),...,c_{L}(\Theta )\right)$$
where $C={\left(c_{1},...,c_{L}\right)}^{T}$ is an *L*-dimensional objective space. The layered coverage curve shown in Figure 4, which corresponds to Figure 2, confirms the necessity of introducing PEM. At different altitudes, the effective coverage rate is significantly different, rather than a fixed 78.5%. The radars within a radar network complement each other’s blind spots and achieve effective coverage at different altitudes, as referred to in Equation (9). The solutions of Equation (8) tend to generate a deployment plan that maximizes the overall coverage in ROI, while the deployment plan suggested by Equation (9) focuses on achieving effective coverage across different altitudes.

### 4.2. LEC-NSGA3

We employ NSGA-III to solve (9), which is a reference-point-based many-objective evolutionary algorithm and follows the NSGA-II framework. The core steps of NSGA-III encompass generating reference points, mutation, crossover, population sorting, and selection.

A preprocessing step is kept consistent for our method, and its purpose is to generate a small set of candidate sites $\left\{\Theta \right\}$ to further reduce computational complexity. This preprocessing can be achieved by gridding the DEM using a non-overlapping sliding window and identifying the highest point in each window as a potential candidate site. This approach can effectively exclude some low-lying positions from the set of candidate sites. In this paper, each individual in the population is represented by the locations of its radars, denoted as ${\left[y_{1},z_{1},...,y_{N},z_{N}\right]}^{T}$. Note that the radar installation height, $\left\{x_{i},i=1,...,N\right\}$, is derived from the terrain data.

The initial population is obtained through random candidate site coordinates. The fitness function is derived from the optimization problem (9). After 200 iterations, the most fit individual can be generated, providing a solution to optimization problems.

The calculation of $P^{net}$ is based on Formula (1), while $P^{i}\in \left\{P^{s}\right\}$ is directly read from the pre-saved data file by Algorithm 1.

Here, a summary of LEC-NSGA3 is provided in Algorithm 2.
**Algorithm 2** LEC-NSGA3**Input:** DEM data and parameters of NSGA-III, such as population size *G* and number of iterations *K*.**Output:** The optimal sites
$\hat{\Theta }$
(1)Set *k* = 0 and initialize the population
$\left\{\Theta_{g}^{k}={\left[\theta_{1,g}^{k},\theta_{2,g}^{k},...,\theta_{N,g}^{k}\right]}^{T},g=1,2,...,G\right\}$.(2)/* NSGA-III Main Loop */
  **for**  
k=1,...,K **do**
   Crossover and mutation;   Calculate joint detection probabilities for all individual by (1) and Algorithm 1;   Sort the population using LECR;   Selection to obtain a new population;  **end for**
(3)Select top-ranked individual as the final solution
${\hat{\Theta }}_{2}$


## 5. Experiments and Discussion

### 5.1. Experiment Setup

We use the same high-resolution DEM data as in [19] that cover a surveillance region of 3001 × 3001 m^2^ in a Chinese city, with a resolution of 1 m and with altitudes between 450 and 700 m. The radar simulation parameters are also the same as in [19], as follows: the carrier frequency is 9.35 GHz, the pulse repetition frequency PRF is 40 kHz, and the bandwidth B is 20 MHz.

In this article, LSS targets are assumed to be the detection target of N, with a typical flight altitude of less than 1 km. The number of radars in the radar network is N=3, and the area corresponding to all terrain data is about 9 km^2^. We set $R_{\max}=1500$ m, $N_{a}=72$, $N_{r}=1500$, and L=250; in other words, the elevation range within the ROI corresponding to the DEM in Figure 5 is from 450 to 3000 m, with an interval of 50 m. After setting the sliding windows size m=50, the total number of candidate sites is S=3721, and they are marked with white circles in Figure 5.

All experimental results were obtained using Matlab 2022b on a PC with Intel i9-10850K CPU and 32 GB RAM.

### 5.2. Comparison of Propagation Models

The different propagation models affect how much valuable information can be extracted from terrain data. When using the same terrain data, the predicted propagation loss results for four propagation models are shown in Figure 6. Obviously, there is no difference in radio wave propagation both above and below the ground surface in Figure 6a. It can be concluded that FS does not take into account the impact of terrain on propagation. The terrain shadowing judged by LoS is either fully present or absent, without nuance. Therefore, from Figure 6b, there is no obvious diffusion phenomenon of radio waves observed in areas far away from the radar. PL considers terrain shadowing and propagation loss, but it exhibits significant shortcomings when r>1000 in Figure 6c. Note that only PEM takes into account the radar’s antenna pattern, and no information on the antenna beamwidth is observed in Figure 6a–c. Furthermore, from Figure 6d, we can observe phenomena such as the overhead blind zone, beam contour, terrain shadowing, refraction, and diffraction.

From the comparative experiments of propagation models, we can conclude that the introduction of PEM is necessary. Unlike in 2D environments, in 3D non-free space, terrain, atmosphere, and antennas all affect radio wave propagation. Only with higher-accuracy propagation predictions can we assess the coverage situation at different altitude layers. The deployment of multi-node small radar networks is often used for filling detection blind spots at low altitudes, making the impact of terrain particularly critical.

### 5.3. Results of Proposed Method

To demonstrate the importance of terrain and propagation models, two comparative experiments are conducted without considering terrain and using FS as the propagation model. In the two comparative experiments, the maximum detection distances $R_{\max}$ are 1200 m and 1500 m, respectively, with other settings remaining the same as mentioned above. Having optimized the maximization of original ECR using NSGA-III, the results are shown in Figure 7. The deployment of radars is similar to the triangular station deployment or diamond deployment, which is a commonly used station deployment strategy in practice [30]. However, based on actual DEM, this deployment of comparative experiments cannot be adopted as the location of radar stations on slopes or rivers is unrealistic.

The effective coverage results of LEC-NSGA3 at different altitude layers are shown in Figure 8. It is evident that lower altitudes are more susceptible to terrain blockage, resulting in poorer effective coverage. Conversely, higher altitudes have a larger blind spot in the radar headspace, making it crucial for radars to compensate for each other’s blind areas. The layered coverage result is significantly different from the comparison results in Figure 7, and it is more in line with reality. This also demonstrates that terrain cannot be ignored in radar deployment issues.

In addition, we do not see large areas of headspace blind zones in Figure 8, but we can observe crescent-shaped blue areas. It can be inferred from Figure 3 that the top-space blind zone of a single radar corresponds to the circular blue area above the radar. These crescent-shaped blue areas in Figure 8 are part of the top-space blind zone of a single radar, and they are not circular because they are compensated by the top-space blind zones of other radars.

To better showcase the superiority of the proposed methods, Figure 9 displays the detection probability profile in the 3D view. The highest detection probability is observed in the overlapping area covered by three radars. In the overlapping area covered by two radars, a distinct detection distance boundary perpendicular to the horizontal direction is visible. The deployment of three radars complements each other, reducing the headspace blind area. Additionally, due to their varying deployment altitudes, the terrain blockage effect is minimized as much as possible. However, unlike the two comparative experiments, it is evident that the radar network cannot fully cover every altitude layer of the surveillance region, when N=3.

### 5.4. Comparison of Different EAs

To validate the generalization ability of the proposed methods for terrain data, 22 GIS data are randomly selected. These digital geographic data files are 7.5-by-7.5 min topographic quadrangle DEM blocks, with each block based on a data spacing of 30-by-30 m. During the experiment, a 9 km^2^ section of each map is intercepted, containing cities, mountains, rivers, and oceans, as shown in Figure 10. After setting the sliding window size m=2, the total number of candidate sites is S=2601. Other settings remain the same as above. Considering the sensitivity of EAS to the initial population, we performed the experiment five times on this set of GIS data and took the average of the LECRs for the candidate results.

To show the optimization process of NSGA-III, the mean curve of LECRs at different generations is shown in Figure 11. Overall, the low-altitude areas corresponding to L<50, which are below the ground surface, cannot be covered, and LECRs are 0%. When L>230, the effective coverage gradually decreases due to the presence of a top-space blind zone. As the number of iterations k increases, the LCER curves in Figure 11 gradually change from black on the left to red on the right. The most significant increase occurs in the range of 150<L<200. This demonstrates the effectiveness of NSGA-III in searching for the optimal deployment for the LEC problem.

The optimizing deployment problem for maximizing LECR is a non-convex optimization problem. Several classic EAs are employed to tackle (9), including GA [31], PSO [32], MOEA/D [15], and NSGA-II [33]. For GA, MOEA/D, and NSGA-II, the population size and number of generations are set to 50 and 200, respectively. For PSO, the swarm size is set to 50, and the number of iterations is 200. Considering the sensitivity of EAs to the initial population, each EA is run five times on this set of GIS data and the mean value of the results is taken.

The LECR curves for different EAs are displayed in Figure 12. Since a higher LECR indicator is preferable, among all the LECR curves, the results of NSGA-III and MOEA/D outperform those of GA, PSO, and NSGA-II. When L<150, the difference in LECR between NSGA-III and MOEA/D is less than 0.2%. Correspondingly, the LECR curve of NSGA-III is overall shifted upwards by two altitude layers. However, when 200<L<230, the LECR curve of MOEA/D performs poorly, even shifting to the left compared to NSGA-II. To display it more clearly, we have zoomed in on the red boxed area. Obviously, in this area, NSGA-III’s LECR is 2% higher than that of other EAs.

## 6. Conclusions

The detection performance of LSS targets under the constraint of known 3D terrain data primarily depends on the radar network deployment scheme. To address this issue, we introduce PEM into RND with a known terrain and propose LECR to evaluate the effectiveness of the deployment. Simulation experimental results demonstrate that both terrain and propagation effects cannot be ignored, and LEC-NSGA3 is effective for DEM and GIS data.

In the future, more accurate propagation models with low computational complexity and efficient high-dimensional multi-objective evolutionary algorithms will be crucial research areas. This is because the proposed method has some limitations. On the one hand, although PEM can provide more accurate predictions of propagation, it has higher computational complexity. On the other hand, compared to a single numerical ECR, LECR offers a clearer reflection of coverage at different altitude layers, but it may result in a higher-dimensional optimization objective.

## Figures and Tables

**Figure 1 sensors-25-01964-f001:**
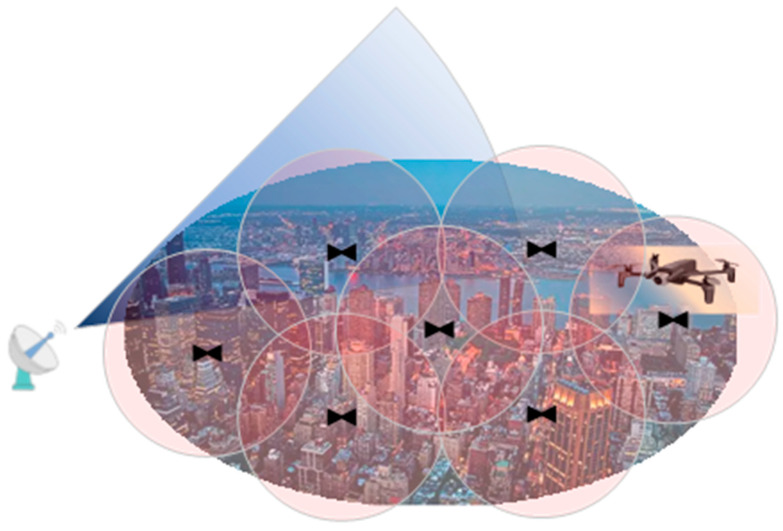
A scene diagram of the joint detection of LSS targets by a multi-node small radar network and other radars. The deployment and setup of the multi-node small radar network are more flexible, making it highly suitable for filling detection blind spots at low altitudes. Additionally, note that terrain shielding must be considered in complex terrain environments.

**Figure 2 sensors-25-01964-f002:**
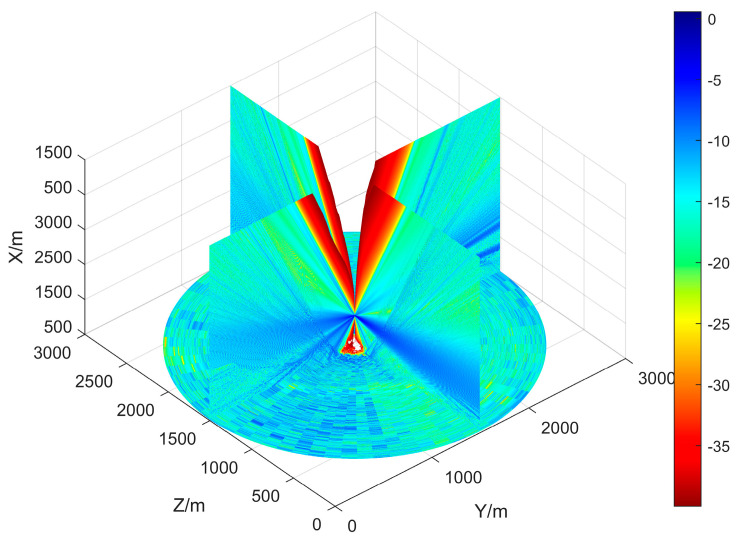
The 3D view of the propagation loss $L_{\mathrm{dB}}^{M}$ of a radar. The propagation losses in the radar’s headspace blind zone, beam strong coverage area, and ground clutter area are clearly distinguished by different colors.

**Figure 3 sensors-25-01964-f003:**
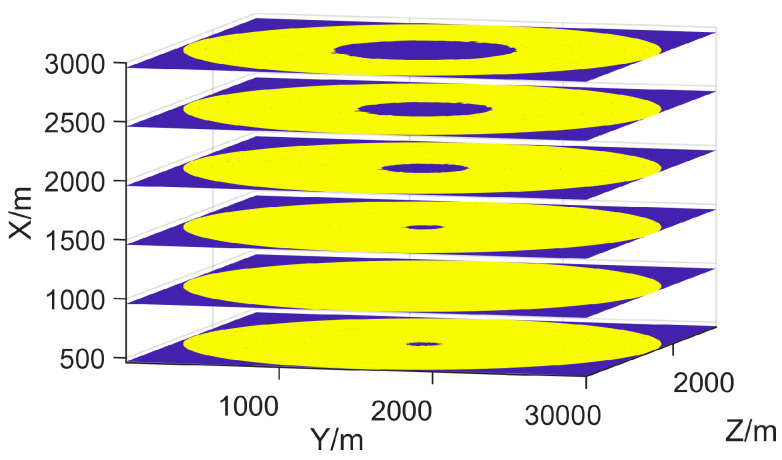
The effective coverages of a radar at different altitudes. Note that the effective coverage area is highlighted in yellow, while other areas are marked in blue. The coverage at different altitude layers is significantly different.

**Figure 4 sensors-25-01964-f004:**
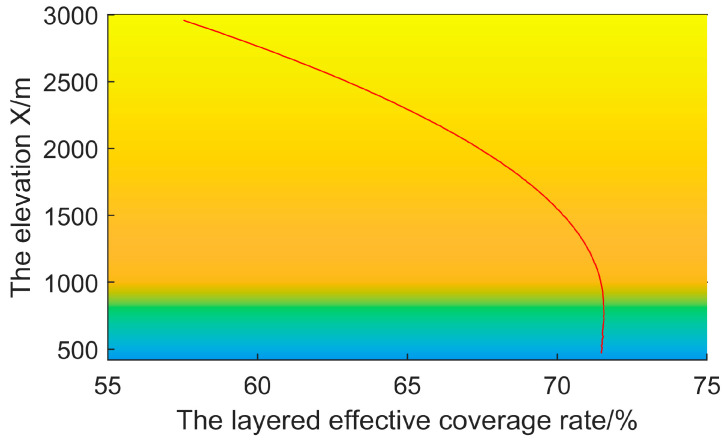
The layered coverage curve of a radar and its corresponding layered coverage curve at different altitudes. The layered coverage curve is divided into two parts, corresponding to the headspace blind zone and the ground clutter zone, respectively.

**Figure 5 sensors-25-01964-f005:**
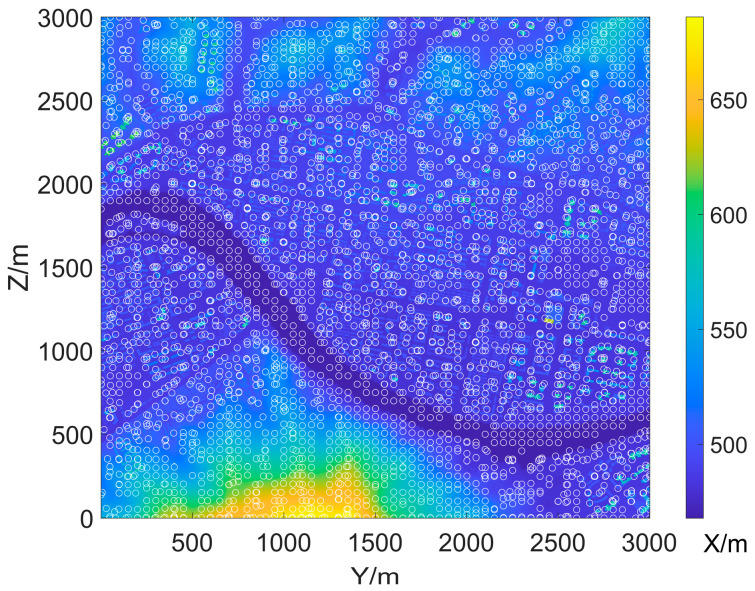
DEM of the surveillance region and candidate sites marked by white circles.

**Figure 6 sensors-25-01964-f006:**
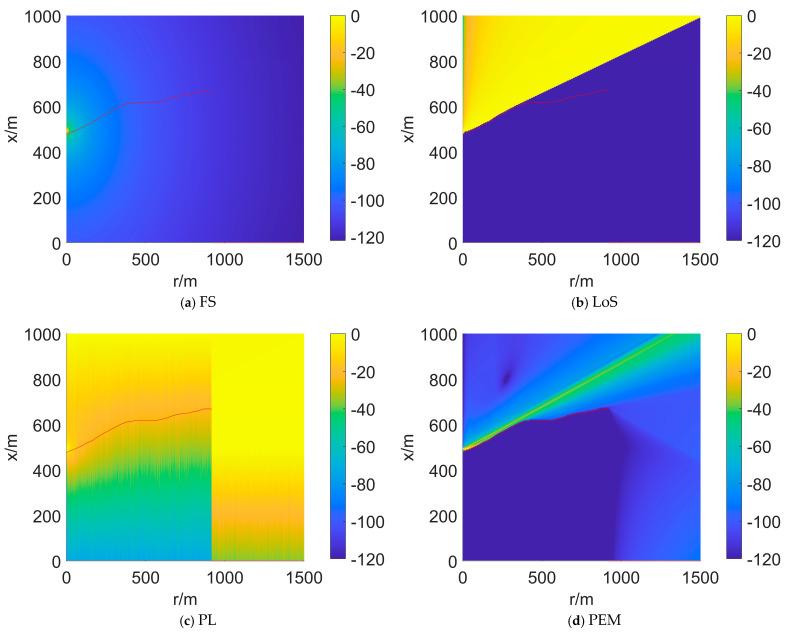
Comparison of the predicted propagation losses from different propagation models. The elevation profile reflecting the terrain is shown as a red dotted line. (**a**–**d**) correspond to FS, Los, PL, and PEM, respectively.

**Figure 7 sensors-25-01964-f007:**
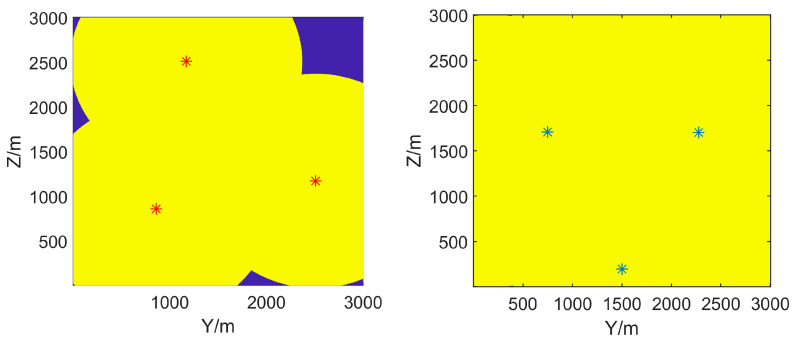
Two results without considering terrain. Note that the effective coverage area is highlighted in yellow, while other areas are marked in blue. The optimal positions are marked with a red and blue *. When $R_{\max}=1200$, the surveillance region cannot be completely covered. But the coverage rate is close to 100% when $R_{\max}=1500$.

**Figure 8 sensors-25-01964-f008:**
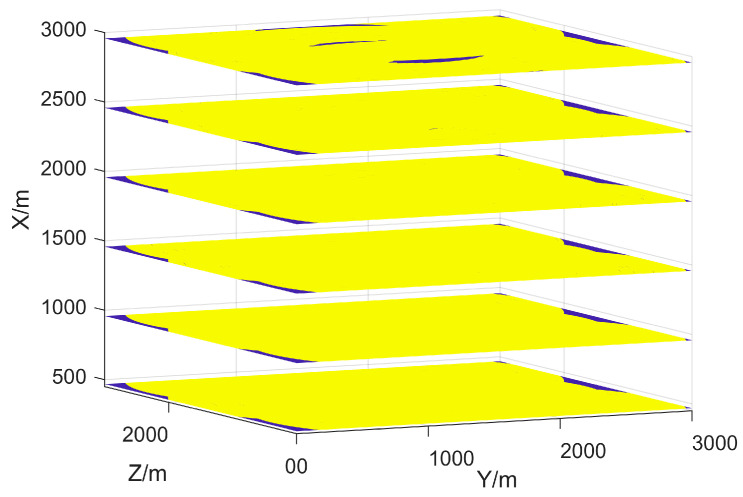
The effective coverage areas of the radar network at different altitudes h = 450~3000 m. The effective coverage areas are marked in yellow, while other areas are marked in blue.

**Figure 9 sensors-25-01964-f009:**
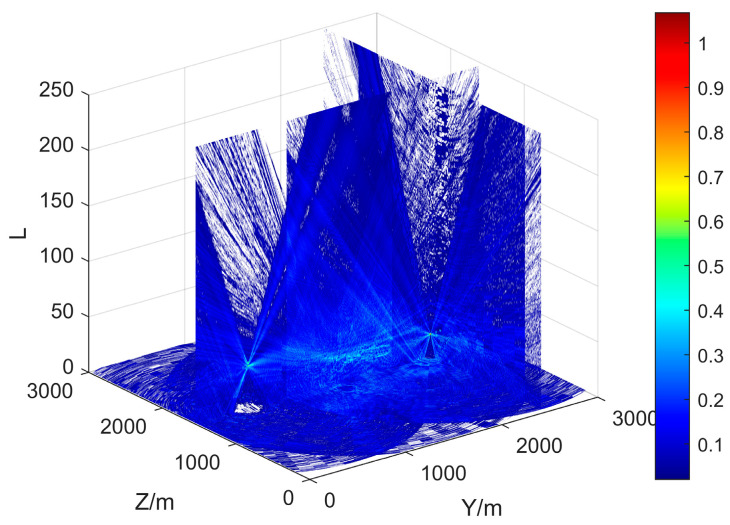
The detection probability $P^{net}$ in a 3D view.

**Figure 10 sensors-25-01964-f010:**
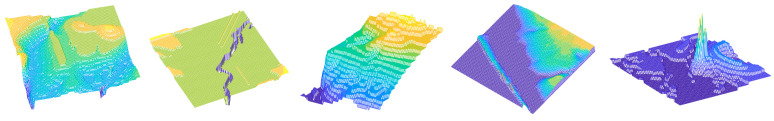
Three-dimensional views of GIS data for various terrains. Different elevations in these figures are marked with different colors.

**Figure 11 sensors-25-01964-f011:**
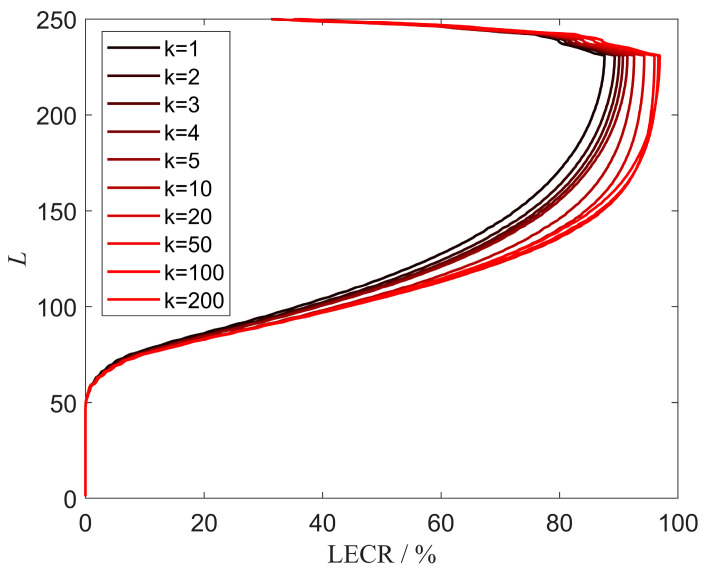
LECR curves results for different numbers of iterations k.

**Figure 12 sensors-25-01964-f012:**
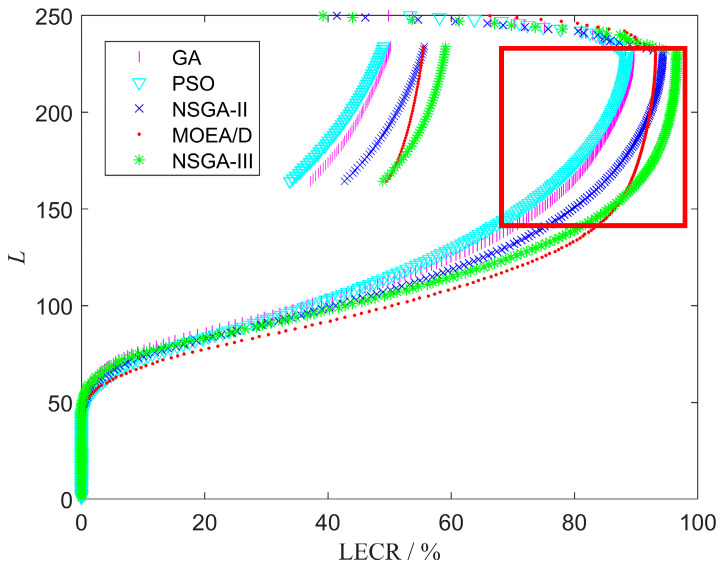
Comparison of LECR results for different EAs on 22 GIS data. The portion of LECR curves with L∈[150,200], circled in red, is enlarged.

**Table 1 sensors-25-01964-t001:** Comparison of different model complexities.

Model	Complexity
FS	$o\left(SN_{a}N_{r}L\right)$
LoS	$o\left(SN_{a}N_{r}LM_{d}\right)$
PL	$o\left(SN_{a}N_{r}LN_{p}\right)$
PEM	$o\left(SN_{a}N_{r}L\log L\right)$

## Data Availability

The data presented in this study are available on request from the corresponding author. The data are not publicly available due to project restrictions.

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
