# Peer review of "Multi-Node Small Radar Network Deployment Optimization in 3D Terrain"

_sensors, 2025, doi:10.3390/s25071964_

Round 1
Reviewer 1 Report
Comments and Suggestions for Authors
This manuscript investigates a practical problem: the deployment of multi-node small radar networks under 3D terrain. The authors consider the propagation of radio waves in areas with known three-dimensional (3D) terrain,introduce the parabolic equation model, establish the Layered effective coverage rate, and use NSGA-III to solve the problem well. However, there are some weaknesses in this manuscript:
- The description in the first paragraph of the Introduction lacks foundation. For example“Traditional radars are not effective in detecting low-altitude slow-moving small (LSS) targets and are not suitable for use in complex environments such as cities and mountains”. Please add citations to strengthen the evidence.
- What are the advantages of small radar over traditional radars in detecting LSS targets as well as in 3D terrain deployment? Please add the evidence. Otherwise, the argument is not sufficient.
- The deployment of radar networks has been extensively researched. For example, DOI: 10.1109/TAES.2023.3347685,DOI: 10.3390/rs15194674,https://doi.org/10.3390/rs16214004,DOI: 1109/taes.2024.3368378, DOI: 10.1016/j.sigpro.2017.09.006,DOI: 10.1109/TAES.2021.3069269,DOI: 10.1049/IET-RSN.2017.0351,DOI: 10.1049/JOE.2019.0188. However, in this manuscript, a detailed review of the deployment of existing radar networks is lacking, which makes the manuscript insufficiently well thought out. The review can be referred to DOI: 10.3390/rs17040730.
- This manuscript lacks a scene diagram, please add.
- There are several notable errors in this manuscript. For example, “Error! Reference source not found” in line 223, the formulas in line 230 and 327 has moved up significantly.
Author Response
Reviewer: 1
Comments:
This manuscript investigates a practical problem: the deployment of multi-node small radar networks under 3D terrain. The authors consider the propagation of radio waves in areas with known three-dimensional (3D) terrain,introduce the parabolic equation model, establish the Layered effective coverage rate, and use NSGA-III to solve the problem well. However, there are some weaknesses in this manuscript:
1: The description in the first paragraph of the Introduction lacks foundation. For example “Traditional radars are not effective in detecting low-altitude slow-moving small (LSS) targets and are not suitable for use in complex environments such as cities and mountains”. Please add citations to strengthen the evidence.
Response: Thank you to the reviewer. There are many methods for detecting low, small, and slow targets, such as multistatic radars, MIMO radars, and more. Due to the characteristics of low, small, and slow targets, traditional radar detection faces issues with ground clutter shielding at low altitudes. Relevant citations and descriptions have been added to the introduction section of this paper. Of course, this is only to introduce the optimization problem of multi-node small radar deployment under terrain constraints, so the related text content has been added minimally. This is also to avoid diluting the research content of this paper, which is radar network deployment.
2: What are the advantages of small radar over traditional radars in detecting LSS targets as well as in 3D terrain deployment? Please add the evidence. Otherwise, the argument is not sufficient.
Response: Thank you to the reviewer for pointing out this issue. I have made some revisions to the introduction section of this paper, adding the advantages of small radars and multi-node small radar networks in addressing LSS (low, small, and slow) targets. Additionally, I have acknowledged that other types of radars and sensors, such as infrared, visible light, laser, and acoustic sensors, are also essential when detecting LSS targets. Considering that the focus of this paper leans towards radar, I have only cited research literature on radar sensors and have not included literature on LSS target detection using other types of sensors. Additionally, I have clearly pointed out the research focus of this paper, which is the optimization of multi-node small radar network deployment under terrain constraints.
3: The deployment of radar networks has been extensively researched. For example, DOI: 10.1109/TAES.2023.3347685,DOI: 10.3390/rs15194674,https://doi.org/10.3390/rs16214004,DOI: 1109/taes.2024.3368378, DOI: 10.1016/j.sigpro.2017.09.006,DOI: 10.1109/TAES.2021.3069269,DOI: 10.1049/IET-RSN.2017.0351,DOI: 10.1049/JOE.2019.0188. However, in this manuscript, a detailed review of the deployment of existing radar networks is lacking, which makes the manuscript insufficiently well thought out. The review can be referred to DOI: 10.3390/rs17040730.
Response: The reviewer mentioned 9 references, which are: (1) DOI: 10.1109/TAES.2023.3347685; (2) DOI: 10.3390/rs15194674; (3) https://doi.org/10.3390/rs16214004; (4) DOI: 1109/taes.2024.3368378; (5) DOI: 10.1016/j.sigpro.2017.09.006; (6) DOI: 10.1109/TAES.2021.3069269; (7) DOI: 10.1049/IET-RSN.2017.0351; (8) DOI: 10.1049/JOE.2019.0188; and (9) DOI: 10.3390/rs17040730. References (1) and (2) are all about radar detection research related to unmanned aerial vehicles or ships. References (3), (5), (7), (8), and (9) all utilized the multi-objective particle swarm optimization algorithm. References (4) and (6) focus on improving the performance of sensor networks. Among them, the research content of some literature involves unmanned aerial vehicle detection, which is consistent with the research content of our paper. However, they are based on MIMO and bistatic radar systems, rather than multi-node small radars. I have added citations of these references in the background introduction section and the algorithm introduction section at the beginning of the paper, to support the introduction of relevant research backgrounds such as unmanned aerial vehicle detection, sensor deployment in complex environments, and optimization algorithms.
4: This manuscript lacks a scene diagram, please add.
Response: Thank you for your suggestions. In the introduction section of this paper, we have incorporated a schematic diagram illustrating the joint detection of LSS (low, small, and slow) targets by a multi-node small radar network in conjunction with other radars. The inclusion of this diagram enhances the explanation of the advantages of multi-node small radars in addressing low-altitude detection blind spots and demonstrates the challenges posed by terrain shielding in radar detection under complex terrains. This underscores the significance of the research presented in this paper.
5: There are several notable errors in this manuscript. For example, “Error! Reference source not found” in line 223, the formulas in line 230 and 327 has moved up significantly.
Response: Thank you to the reviewer for pointing out the formatting issues. There was a problem with formula citations on line 223, which resulted in an error message. Thanks to the reviewer's thorough check, we have corrected these formatting issues. After carefully examining all the formulas in the paper, we also found some formula issues with their sizes and have made the necessary corrections.

Reviewer 2 Report
Comments and Suggestions for Authors
The manuscript is well constructed. Though the study is outside the scope of my field, I believe that after necessary revision, it is ready for publication. My comments and suggestions are as follows:
1) Limitations of the proposed method must be concluded.
2) More references should be cited to present a more detailed state-of-the-art of previous studies.
3) Figure 5a-c is not interpreted in the main text.
4) In Figure 2, is the blue region in the yellow region blind zone? Why did the coverage rate decrease when the elevation is less than 1000 m?
5) Does the headspace blind zone influence the results of the proposed method?
6) Figure 10 and Figure 11 are not interpreted in the main text. The authors only present the results, please comment on the comparisons.
Author Response
Reviewer: 2
Comments:
The manuscript is well constructed. Though the study is outside the scope of my field, I believe that after necessary revision, it is ready for publication. My comments and suggestions are as follows:
- Limitations of the proposed method must be concluded.
Response: Yes, the proposed method has its limitations. On the one hand, the deployment of multi-node small radar networks is mainly targeted at LSS targets. For targets flying at higher altitudes, terrain constraints may be negligible. Furthermore, while introducing a better propagation model can improve the accuracy of propagation predictions, it also increases computational complexity. Of course, this paper is the first to introduce PEM into the radar deployment optimization problem. This is because when deploying multi-node small radar networks, terrain data is usually relatively easy to obtain, and the impact of terrain on propagation cannot be ignored. On the other hand, the layered effective coverage rate, as an optimization objective, is a high-dimensional and multi-objective problem. This paper adopts NSGA-III for searching and solving, which is the result of comparisons among different evolutionary algorithms. Currently, state-of-the-art multi-objective evolutionary algorithms are mainly targeted at objectives up to a dozen dimensions. Therefore, we believe that developing better high-dimensional and multi-objective optimization algorithms will be an important direction for future research.
Addressing these limitations, we have added a description of the limitations of the proposed algorithm in the conclusion section of the paper, outlining directions for future research.
- More references should be cited to present a more detailed state-of-the-art of previous studies.
Response: Thank you for your suggestions. In the introduction section of the paper, we have added nine related studies, mainly including research on sensor deployment in complex environments, research on low, small, and slow target detection performance as an optimization objective, and different types of radars (such as MIMO radar systems, multistatic radars, etc.). Obviously, there is a vast amount of related research, and it is not possible for us to cite all of it. However, we have categorized the related research into three aspects: propagation models, optimization objectives, and evolutionary algorithms. The references we have cited can also be roughly classified into these three categories, and the relevant text has been added to the introduction section of the paper. If you believe there are other references that need to be cited but have not been included, please let us know, and we will make the necessary modifications.
- Figure 5a-c is not interpreted in the main text.
Response: Yes, the relevant explanations were omitted previously. In Section 5.2, we have added interpretations for these figures. The added text mainly describes the details of these figures and emphasizes the advantage of PEM compared to other propagation models—its ability to consider the impacts of atmosphere and antennas on propagation. Additionally, we have separately added a paragraph to draw the conclusion of this experiment—the introduction of PEM is necessary. The deployment of multi-node small radar networks is often used for filling detection blind spots at low altitudes, making the impact of terrain particularly critical.
- In Figure 2, is the blue region in the yellow region blind zone? Why did the coverage rate decrease when the elevation is less than 1000 m?
Response: Yes, effective coverage is binary. In Figure 3 (previously Figure 2 due to the addition of a new figure in the introduction section), yellow represents areas with effective coverage, while blue represents areas without effective coverage. The yellow areas are mainly concentrated around the radar, so the periphery of these yellow areas mainly consists of regions beyond the radar's detection range, hence they are colored blue. At the same time, there are also blue areas within the yellow regions. For example, at altitude layers above the radar installation height (at an altitude of 1000m), the blue areas within these altitude layers mainly correspond to the radar's top-space blind zone. Of course, when the altitude is below 1000m, there are also blue areas within the yellow regions, mainly due to the ineffective coverage caused by strong ground clutter. The top-space blind zone gradually expands as altitude increases. Therefore, in Figure 3, when the altitude is above 1000m, the blue areas within the yellow regions gradually enlarge. The strong ground clutter area is smallest at the radar installation altitude of 1000m. When the altitude is below 1000m and continues to decrease, obstructions from the terrain generate strong ground clutter. The enhancement of ground clutter leads to a decrease in the signal-to-clutter ratio, resulting in an increase in the area of ineffective coverage (blue areas).
In addition, Figures 3, 2, and 4 are interrelated. First, the propagation loss map (Figure 2) is obtained using PEM, and then the layered effective coverage map (Figure 3) is derived from it. The effective coverage rate differs at various altitude layers, and these are plotted as a layered effective coverage rate curve to obtain Figure 4. From the section of Figure 4 where the altitude is below 1000m, it can be observed that the effective coverage rate curve has a tendency to shift to the left, indicating a decreasing trend in LECR. The reason for this decrease is the strong ground clutter mentioned above.
- Does the headspace blind zone influence the results of the proposed method?
Response: The method proposed in this paper can effectively address the impact of the top-space blind zone. It should be noted that due to the introduction of PEM, the obtained layered effective coverage maps (Figures 3 and 8) include the top-space blind zone. Figure 3 shows the layered effective coverage of a single radar, so it is clear that the top-space blind zone exists above the radar and its area gradually increases. Figure 8 presents the layered effective coverage of three radars. At the highest altitude layer of 3000m, we do not see large areas of headspace blind zone, but we can observe crescent-shaped blue areas. These crescent-shaped blue areas are part of the top-space blind zone of a single radar, and they are not circular because they are compensated by the top-space blind zones of other radars.
Thank you for your reminder. We have added some explanatory text for Figure 8 in Section 5.3, emphasizing that the crescent-shaped blue areas differ from the circular blue areas in Figure 3.
- Figure 10 and Figure 11 are not interpreted in the main text. The authors only present the results, please comment on the comparisons.
Response: Thank you for your thorough check. We have also noticed that Figures 11 (previously Figure 10, due to the addition of a new figure in the introduction section) and 12 (previously Figure 11, now Figure 12) lacked corresponding explanatory text in the main text. We have added relevant explanatory paragraphs at the corresponding positions in Section 5.4. Additionally, there was an issue with the title of Figure 11, which has now been corrected. The resolution of these two figures was also low, so we have updated them. The purposes of these two figures are to demonstrate the effectiveness and superiority of the proposed algorithm, respectively, and this content is now clearly described in the main text.

Reviewer 3 Report
Comments and Suggestions for Authors
In this paper, the parabolic equation model is introduced to radar network deployment considering terrain constraints. The proposed nondominated sorting genetic algorithm is to solve the optimization as Layered Effective Coverage Rate the objectives. Some comments are presented as follows.
- At the beginning of Section 2, the authors are suggested showing the whole optimization problem before entering the detail.
- How is "Split-step Fourier Transform" used in the proposed scheme?
- The computational complexity of the algorithm 1 is cubic. How to decrease the complexity?
- In page 7, the authors should explain "the layered effective coverage problem is a MOP" in detail.
- How do the authors explain Figure 4?
- Figure 10 and Figure 11 should be explained more in terms of performance gain.
Author Response
Reviewer: 3
Comments:
In this paper, the parabolic equation model is introduced to radar network deployment considering terrain constraints. The proposed nondominated sorting genetic algorithm is to solve the optimization as Layered Effective Coverage Rate the objectives. Some comments are presented as follows.
- At the beginning of Section 2, the authors are suggested showing the whole optimization problem before entering the detail.
Response: Thank you for your suggestion. It's a great idea. I believe that even in the field of radar, many people are not familiar with multi-node small radar networks. We have added some text in Section 2 of this paper to briefly describe the tasks and characteristics of multi-node small radar networks. Multi-node small radar networks are mainly used for low-altitude blind spot compensation, and when the type, number, and parameters of the radars are determined, their detection performance is mainly determined by the deployment scheme. After adding the description of the optimization problem, the paper flows more smoothly and will be easier for readers to understand. Additionally, inspired by your suggestion, we have also added background information in the introduction section of the paper.
- How is "Split-step Fourier Transform" used in the proposed scheme?
Response: The Split-step Fourier Transform (SSFT) is a classic algorithm for numerically solving PEM, with well-developed theoretical research and extensive engineering applications. The algorithm proposed in this paper uses PEM as the propagation model and employs SSFT for step-by-step solutions.
Firstly, it should be noted that our Region of Interest (ROI) is 3D, encompassing three dimensions in the Cartesian coordinate system: x, y, and z, where x represents altitude. The radar perspective employs spherical coordinates, also with three dimensions: r, θ, and φ, where r denotes distance. PEM modeling for propagation occurs on a 2D plane, corresponding to the azimuth angle φ of the radar beam, as in u(x, r) in formula (5), which lies in an x-r plane where r is also specified as the direction of wave propagation, with the corresponding beam elevation angle being θ. As mentioned above, within the current azimuth plane φ, SSFT can solve for u(x, r) on that plane. The entire 3D ROI comprises multiple planes with different φ values, corresponding to different azimuth angles scanned by the radar beam. In this way, multiple u(x, r) values are obtained and stitched together to form u(x, r, φ) for the entire ROI. The above describes the entire process of solving PEM using SSFT, corresponding to the "for φ=0~360" loop in Algorithm 1. The next step is a detailed operation: coordinate mapping is performed on u(x, r, φ) to convert it into u(x, y, z) in the Cartesian coordinate system. The subsequent algorithm steps follow Algorithms 1 and 2 in this paper.
Additionally, the SSFT we employed is the wide-angle SSFT from reference [24]. For further reading, yuo can also refer to "Parabolic Equation Methods for Electromagnetic Wave Propagation" by Levy Mireille. The "for φ=0~360" loop in Algorithm 1 simulates the radar beam scanning process, which is done in a similar manner as described in reference [21]. Another reference for this approach is DOI: 10.1109/TAP.2008.2007392, titled "Simulation and Prediction of Weather Radar Clutter Using a Wave Propagator on High Resolution NWP Data".
- The computational complexity of the algorithm 1 is cubic. How to decrease the complexity?
Response: Yes, the computational complexity of Algorithm 1 is high. As mentioned in Section 3.3, the computational complexity of PEM is generally the highest among all propagation models. From the perspective of maintaining radar resolution without reduction, the step size ∆r should ideally not be less than the radar's range resolution during the PEM solution process. Additionally, from the perspective of not reducing the resolution of terrain data, the step size ∆r is also constrained. However, when excessive detail is not required, the step size ∆r can be appropriately increased. The selection of Na and L is similar to the selection of the step size ∆r. By choosing a larger step size ∆r and smaller Na and L, the computational complexity of PEM can be significantly reduced. However, during the experiment, we recognized the need to maintain the resolution of both radar and terrain data and did not make any compromising choices. We performed the PEM calculations offline and saved the single-radar detection probability data for 3721 candidate sites with high-resolution DEM data. During the optimization process, we only needed to read the corresponding data. Therefore, the size of the saved data was important, and we saved the detection probability values after retaining only a few effective digits. This approach effectively accelerated the algorithm's running speed. Of course, having a more efficient and higher-precision propagation model would be better, and this is one of our directions for future improvement.
- In page 7, the authors should explain "the layered effective coverage problem is a MOP" in detail.
Response: Thank you for your suggestions. We have made revisions to the relevant descriptions on page 7. Since maximizing layered effective coverage (LEC) involves maximizing the ECRs across L altitude layers, maximizing LEC is a multi-objective optimization problem (MOP). As in formula (9), the dimension of C is L.
- How do the authors explain Figure 4?
Response: Figure 5 (previously Figure 4, as we have added a new figure in the introduction section) is a two-dimensional representation of high-resolution DEM data, where the colors indicate altitude levels, as shown in the colorbar. Each pixel in this DEM data has a resolution of 1 meter, and the entire data covers an area of 3 by 3 square kilometers. The data was obtained from a remote sensing survey of a city in China conducted by a certain research institute. Figure 5 contains many small white circles, marking 3721 candidate sites. In other words, for the location of radar installation, there are 61 degrees of freedom along each 3km side of the figure. Among these 3721 candidate sites, our experiment requires selecting 3 sites, resulting in a total of 5.1520×1010 candidate solutions. Our task is to choose the best possible deployment scheme from this vast solution set.
- Figure 10 and Figure 11 should be explained more in terms of performance gain.
Response: Thank you for your thorough check. We have also noticed that Figures 11 (previously Figure 10, due to the addition of a new figure in the introduction section) and 12 (previously Figure 11, now Figure 12) lacked corresponding explanatory text in the main text. We have added relevant explanatory paragraphs at the corresponding positions in Section 5.4. Additionally, there was an issue with the title of Figure 11, which has now been corrected. The resolution of these two figures was also low, so we have updated them. The purposes of these two figures are to demonstrate the effectiveness and superiority of the proposed algorithm, respectively, and this content is now clearly described in the main text.

Round 2
Reviewer 1 Report
Comments and Suggestions for Authors
The authors have addressed all my concerns. Thanks for the efforts. The manuscript could be published.
Reviewer 2 Report
Comments and Suggestions for Authors
No more comments.
Reviewer 3 Report
Comments and Suggestions for Authors
The authors have revised the manuscript based on the reviewers' comments.